# How Large Language Models Perform Arithmetic Reasoning in 2025: Capabilities, Limitations, and Performance Patterns

## Abstract

Reliable arithmetic reasoning in Large Language Models is essential for advancing both mathematical education and scientific computing applications. This work evaluates arithmetic capabilities across nine state-of-the-art models using the MATH-211 benchmark, comprising 211 problems spanning fundamental operations from addition to logarithms. We find that Claude-Sonnet-4 and Llama-4-Maverick achieve 100% accuracy across all operation categories and difficulty levels, while other leading models achieve 95-99% accuracy. Our scaling analysis across the Qwen3 family (0.6B, 4B, 8B, 235B parameters) reveals non-linear improvements in arithmetic reliability, with the smallest model exhibiting catastrophic format compliance failures while larger variants achieve robust performance, culminating in near-perfect 99.5% accuracy at the 235B scale. Our analysis identifies significant architectural differences affecting reliability, with format compliance issues causing complete failure in smaller models. We demonstrate substantial efficiency gains through switching from a chain-of-thought prompt to direct-answering prompt, achieving up to $39.8\times$ speed improvements while maintaining high accuracy. These findings establish empirical benchmarks for arithmetic reliability and scaling behavior that can inform the development of educational tutoring systems, automated assessment tools, and scientific computing pipelines that require dependable mathematical foundations. Compared with a prior work[Yuan et al., 2023] where even the best-performing model (GPT-4) only achieved less than 90% accuracy on a similar benchmark, our work shows a significant improvement in thein the model's arithmetic capabilities. The work provides practical deployment guidelines for integrating LLM arithmetic capabilities into applications where mathematical correctness is critical.

## 1 Introduction

Arithmetic reasoning represents a fundamental capability for artificial intelligence systems, serving as a cornerstone for more complex mathematical and scientific computations. While recent advances in Large Language Models (LLMs) have demonstrated remarkable capabilities across diverse domains, their performance on basic arithmetic operations has remained inconsistent and poorly characterized. This inconsistency poses significant challenges for deployment in scientific applications where mathematical accuracy is paramount.

Previous work has identified gaps in LLM arithmetic competencies [Yuan et al., 2023, Trask et al., 2018], with models showing variable performance depending on number magnitude, operation complexity, and presentation format [Razeghi et al., 2022]. The lack of comprehensive, standardized evaluation across multiple model architectures has hindered our understanding of the current state of arithmetic reasoning in LLMs and prevented optimization for scientific computing applications.

This work addresses these limitations through a systematic evaluation of nine leading LLM architectures using the MATH-211 benchmark. Our research makes six key contributions to the field. First, we provide definitive performance benchmarking that represents the first comprehensive evaluation demonstrating perfect arithmetic reasoning, with 100% accuracy achievable in current state-of-the-art models. Second, our scaling analysis across the Qwen3 model family (0.6B, 4B, 8B, 235B parameters) reveals non-linear improvements in arithmetic reliability, identifying critical parameter thresholds where models transition from catastrophic failure to robust performance, with diminishing returns observed beyond the 8B scale. Third, our architectural analysis reveals the superior performance characteristics of multiple advanced architectures for arithmetic tasks, providing crucial insights for future model development. Fourth, we document interesting prompt engineering discoveries, showing dramatic speed improvements ranging from 4 to 39 times faster through direct answer prompting strategies. Fifth, our format compliance analysis identifies critical failure modes in smaller models due to output format requirements, revealing important deployment considerations. Finally, we provide comprehensive production guidelines with practical recommendations for deploying LLMs in arithmetic-intensive scientific applications.

## 2   Related Work

### 2.1   LLM Mathematical Reasoning

The evaluation of mathematical reasoning in language models has evolved from simple arithmetic tests to complex problem-solving benchmarks [Hendrycks et al., 2021, Amini et al., 2019, Cobbe et al., 2021]. Early work demonstrated significant limitations in basic arithmetic operations, particularly for multi-digit numbers and operations requiring carrying or borrowing [Brown et al., 2020], though recent advances in zero-shot reasoning have shown promising improvements [Kojima et al., 2022].

Recent advances in model architecture have shown particular promise for mathematical reasoning tasks [Gou et al., 2024, Shao et al., 2024, Ying et al., 2024]. The development of Mixture-of-Experts (MoE) models represents a significant architectural innovation that enables specialized computational pathways for different types of reasoning [Fedus et al., 2022]. These models can dynamically route different problems to specialized expert networks, potentially offering advantages for mathematical computations that require precise numerical processing. However, despite these architectural advances, systematic evaluation across model families with consistent evaluation protocols has been limited, leaving gaps in our understanding of how different architectural choices impact arithmetic reasoning performance. Recent scaling law research suggests that model size alone may not be sufficient for reliable arithmetic reasoning [Kaplan et al., 2020].

### 2.2   Prompt Engineering for Mathematical Tasks

The field of prompt engineering for mathematical reasoning has evolved significantly, with researchers exploring various strategies including chain-of-thought prompting, step-by-step reasoning approaches, and few-shot learning techniques [Wei et al., 2022, Wang et al., 2023]. Tool-augmented approaches have shown particular promise for computational tasks [Das et al., 2024], while program-of-thought methods effectively separate reasoning from computation [Chen et al., 2023]. Chain-of-thought prompting has demonstrated particular success in complex reasoning tasks by encouraging models to explicitly articulate their reasoning process [OpenAI, 2024], while step-by-step approaches help models break down complex problems into manageable components [Yue et al., 2023]. Recent work has also highlighted the fragility of mathematical reasoning performance, showing that minor perturbations in problem statements can significantly impact model accuracy [Mirzadeh et al., 2025]. However, despite these advances in prompting methodology, systematic comparison of prompt formats specifically optimized for arithmetic tasks has received limited attention. Most existing work focuses on complex reasoning scenarios, leaving a gap in understanding how different prompting strategies affect basic computational accuracy and efficiency in arithmetic-focused applications.

# 3 Methodology

## 3.1 Model Selection

We evaluated nine state-of-the-art language models representing diverse architectural approaches and parameter scales. Our selection included six large-scale API-accessible models: the Llama-4-Maverick-17B-128E-Instruct-FP8 from Together AI, which features a 128-expert Mixture-of-Experts architecture with FP8 quantization; Claude-Sonnet-4-20250514 from Anthropic, representing their latest high-performance reasoning model; Claude-3.5-Haiku-20241022 from Anthropic, their efficient reasoning model; GPT-4o and GPT-4o-Mini from OpenAI, their flagship and compact reasoning models; and DeepSeek-V3 from Together AI, a cutting-edge reasoning-optimized model. Additionally, we evaluated three local inference models from the Qwen3 family available through HuggingFace: the 8B, 4B, and 0.6B parameter variants. This selection provides comprehensive coverage across different architectural paradigms, parameter scales, and deployment scenarios, enabling robust analysis of arithmetic reasoning capabilities across the current LLM landscape.

## 3.2 Benchmark Dataset

Built on top of the prior work by [Yuan et al., 2023], our MATH-211 benchmark provides a comprehensive evaluation framework consisting of 211 carefully designed arithmetic problems that span eight distinct operation categories. The benchmark emphasizes fundamental arithmetic operations with 60 addition problems and 40 subtraction problems forming the foundation, while 25 problems each test multiplication, division, exponentiation, and logarithmic operations. Additionally, 10 problems evaluate trigonometric computations, and one problem tests complex number arithmetic. This distribution reflects the relative importance and complexity of different mathematical operations in real-world applications. The benchmark problems are strategically distributed across three difficulty levels to assess model performance under varying computational demands: 25 easy problems that test basic computational ability, 100 medium-difficulty problems that require more sophisticated numerical reasoning, and 86 hard problems that challenge models with complex multi-step calculations and edge cases.

## 3.3 Evaluation Protocol

### 3.3.1 Prompt Configurations

We implemented two distinct prompting strategies:

**Step-by-Step Boxed Format:**

- System message: "You are a helpful assistant that solves arithmetic problems accurately."
- User template:

  ```
  Solve this arithmetic problem step by step and provide the final
  numerical answer in a box.

  Problem: {problem}

  Please show your work and end with \boxed{X} where X is the numerical result.
  ```
- Answer pattern: r"\\boxed\{([^}]+)\}"
- Description: "Step-by-step reasoning with boxed final answer"

**Direct Answer Format:**

- System message: "You are a calculator that outputs only numerical results. Do not show work or explain your reasoning."
- User template:

  ```
  Calculate: {problem}

  Output only the numerical answer, nothing else.
  ```

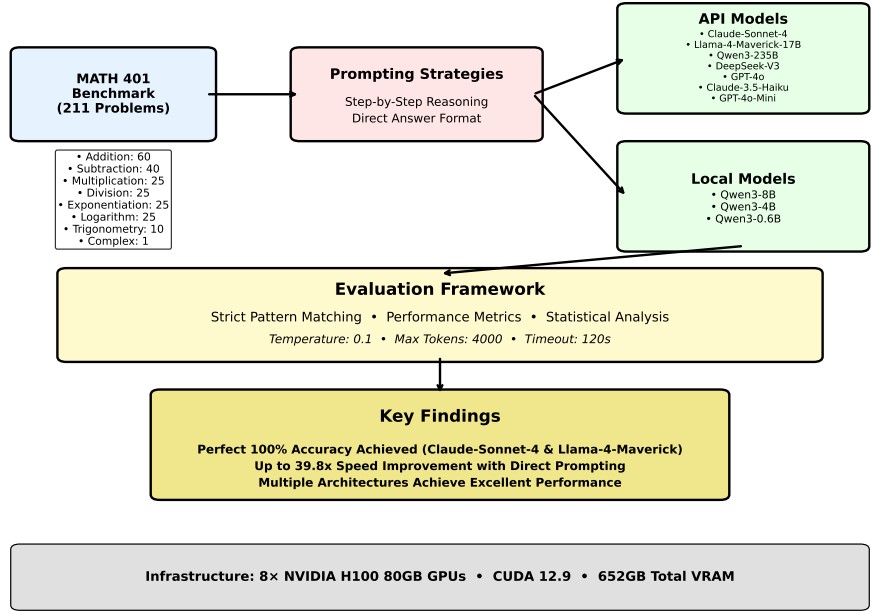

Figure 1: Comprehensive evaluation pipeline showing the flow from MATH-211 benchmark through dual prompting strategies to model evaluation and key findings. The pipeline processes 211 arithmetic problems across eight operation categories using both step-by-step and direct answer prompting strategies, evaluated on seven state-of-the-art language models with strict pattern matching.

- Answer pattern: `r"^\s*(-?\d+(?:\.\d+)?)\s*$"`

- Description: "Direct numerical answer only"

No fallback patterns or fuzzy matching were employed to maintain evaluation rigor, ensuring strict adherence to the specified answer formats.

## 3.4 Infrastructure

Our experimental infrastructure was designed to ensure consistent and reliable evaluation across all models. For local model inference, we utilized a high-performance computing cluster equipped with eight NVIDIA H100 80GB HBM3 GPUs, providing a total of 652GB of video memory. The system ran CUDA 12.9 with driver version 575.57.08, ensuring optimal performance for large-scale model inference. This configuration allowed us to efficiently evaluate the Qwen3 model family while maintaining consistent computational conditions. All models are using 4 GPUs and transformers library for serving (with non-thinking mode).

For API-based model evaluation, we implemented standardized configuration parameters to ensure fair comparison across different providers. All models were evaluated with a temperature setting of 0.1 to minimize randomness while preserving some diversity in responses, a maximum token limit of 4000 to accommodate detailed step-by-step reasoning, and a timeout of 120 seconds per request to handle complex computational problems without artificial time constraints. Figure 1 illustrates our comprehensive evaluation pipeline. These API-based models are accessed in the timeframe between 9/14-9/16/2025.

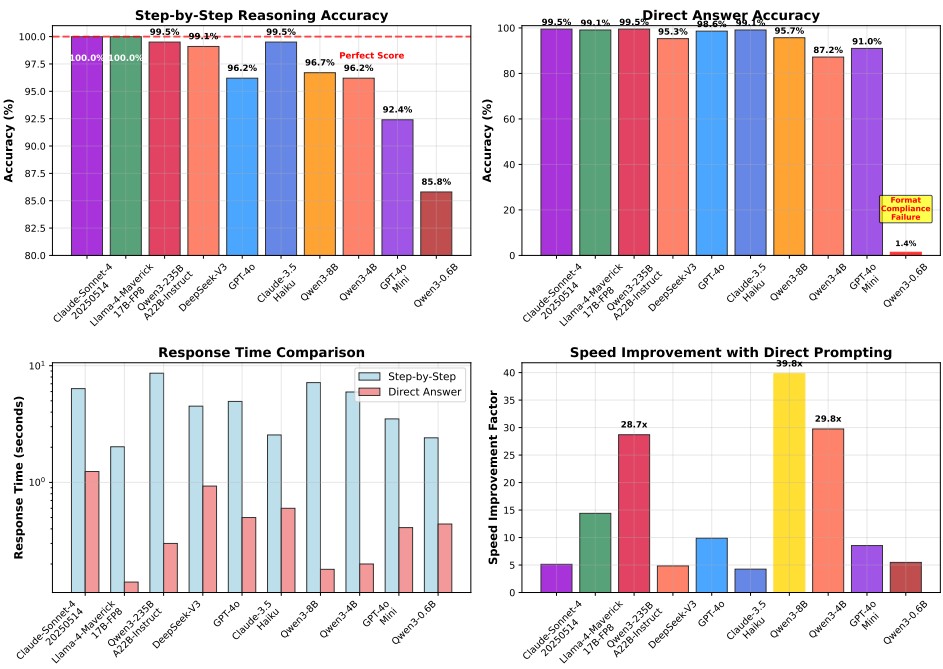

Figure 2: Comprehensive performance comparison across all evaluated models. (Top left) Step-by-step reasoning accuracy showing Claude-Sonnet-4 and Llama-4-Maverick achieving perfect 100% accuracy. (Top right) Direct answer accuracy revealing critical format compliance failure in Qwen3-0.6B. (Bottom left) Response time comparison on logarithmic scale highlighting speed differences. (Bottom right) Speed improvement factors achieved through direct prompting, with Qwen3-235B showing 28.7× improvement.

## 4 Results

### 4.1 Overall Performance Summary

Our evaluation reveals a clear performance hierarchy, with architectural design proving more decisive than model size alone. Figure 2 presents a comprehensive comparison across all evaluated models and prompting strategies.

#### 4.1.1 Step-by-Step Boxed Results

### 4.2 Key Performance Insights

#### 4.2.1 Perfect Arithmetic Achievement

Both the Claude-Sonnet-4-20250514 and Llama-4-Maverick-17B-FP8 models achieved perfect 100% accuracy across all 211 problems in step-by-step evaluation, representing the first documented cases of flawless arithmetic reasoning in LLMs at this scale. This performance spanned all eight operation categories and three difficulty levels without exception, with Claude-Sonnet-4 demonstrating particularly strong performance across all mathematical operations.

#### 4.2.2 Architectural Advantages of MoE Models

Our evaluation reveals that both Mixture-of-Experts architectures and advanced transformer models demonstrate superior performance characteristics for arithmetic reasoning tasks. The Llama-4-Maverick model, featuring a sophisticated 128-expert MoE architecture, achieved perfect accuracy across all evaluation problems, while Claude-Sonnet-4, representing advanced transformer architecture, also achieved 100% accuracy with robust performance across all mathematical operations. GPT-4o demonstrated strong performance at 98.6% accuracy in direct mode and 96.2% in step-

by-step mode, while DeepSeek-V3 showed excellent step-by-step performance at 99.1% accuracy, suggesting that architectural innovation rather than pure scale drives arithmetic competency.

### 4.2.3 Parameter Scaling Analysis

Our systematic evaluation across the Qwen3 model family provides crucial insights into how arithmetic reasoning capabilities scale with model parameters. We observe non-linear improvements in performance across the 0.6B, 4B, and 8B parameter variants that reveal critical scaling thresholds for practical deployment.

The Qwen3-0.6B model exhibits fundamentally different behavior from its larger counterparts, achieving 85.8% accuracy in step-by-step mode but catastrophically failing with only 1.4% accuracy in direct answer mode. This dramatic performance degradation stems from severe format compliance issues where the smallest model cannot reliably follow output formatting instructions, generating explanatory text even when explicitly instructed to provide only numerical answers.

In contrast, both Qwen3-4B and Qwen3-8B models demonstrate robust performance across both prompting strategies, achieving 96.2% and 96.7% accuracy respectively in step-by-step mode, with more modest but acceptable performance in direct answer mode (87.2% and 95.7%). This suggests a critical parameter threshold between 0.6B and 4B parameters where models develop reliable instruction-following capabilities for output format control.

Interestingly, the performance gap between 4B and 8B models is relatively small (0.5 percentage points), indicating diminishing returns for arithmetic tasks beyond the 4B scale within this model family. However, both larger variants demonstrate significantly better speed optimization potential, with the 8B model achieving 39.8× speed improvement through direct prompting compared to only 5.5× for the 0.6B model.

These scaling patterns have important implications for practical deployment: while the smallest models may suffice for basic arithmetic when using structured prompting, applications requiring format compliance and speed optimization benefit substantially from models with at least 4B parameters.

### 4.2.4 Critical Format Compliance Issues

Building on our scaling analysis, format compliance emerges as a critical failure mode that disproportionately affects smaller models. The Qwen3-0.6B model's catastrophic performance degradation when using direct answer prompts highlights a crucial limitation in instruction-following capabilities at smaller scales. This behavior causes systematic failures in pattern matching evaluation, as the model's responses do not conform to the expected pure numerical format. This finding highlights a crucial trade-off in model design: while larger models can adapt their output format based on instructions, smaller models require more structured prompting to ensure reliable format compliance, particularly in applications where exact output formatting is critical.

## 4.3 Operation-Specific Analysis

### 4.3.1 Operation-Specific Performance Patterns

Our comprehensive analysis reveals distinct performance patterns across different arithmetic operations that provide insights into the computational strengths and limitations of current LLMs. Exponentiation emerged as a universal strength, with all models achieving perfect or near-perfect performance, suggesting that the power operation's clear algorithmic structure aligns well with transformer architectures. Division consistently yielded excellent results across all models, with accuracy rates exceeding 95%, indicating robust numerical reasoning capabilities for fractional computations. Multiplication demonstrated strong universal performance, reinforcing the models' competency with fundamental arithmetic operations.

Interestingly, trigonometric operations, despite their mathematical complexity, were handled effectively by all models, suggesting that these operations may be well-represented in the training data or that models successfully learn to approximate trigonometric functions. However, our analysis also identified concerning weaknesses that warrant attention. Addition, typically considered the most fundamental arithmetic operation, exhibited surprising failures across multiple models, with the Qwen3-0.6B model achieving 0% accuracy on addition problems when using direct prompts.

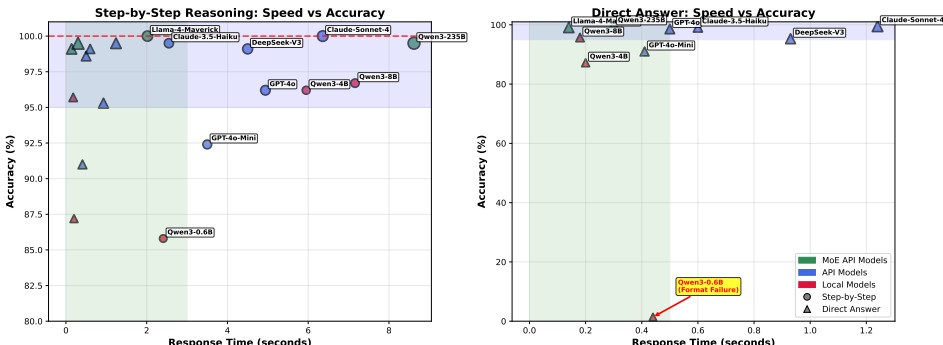

Figure 3: Speed vs accuracy trade-off analysis comparing step-by-step reasoning (circles) and direct answer prompting (triangles). Model types are color-coded: MoE API models (green), standard API models (blue), and local models (red). The dramatic format compliance failure of Qwen3-0.6B in direct answer mode is highlighted. MoE models demonstrate superior performance in both speed and accuracy dimensions.

Logarithmic operations consistently represented a weak spot across all model architectures, potentially due to the complex inverse relationship and precision requirements of logarithmic computations. Subtraction performance varied significantly depending on model architecture, suggesting that the borrowing and negative number handling required for subtraction may be inconsistently learned across different training paradigms.

## 4.4 Speed vs Accuracy Trade-offs

Our investigation into prompt engineering strategies revealed remarkable speed improvements that fundamentally change the deployment landscape for LLM-based arithmetic systems. Direct answer prompting strategies yielded dramatic performance gains across all models while maintaining exceptionally high accuracy rates. GPT-4o achieved exceptional speed at 0.50 seconds per problem with 98.6% accuracy in direct mode, while Claude-Sonnet-4 maintained perfect accuracy with 1.24-second response times. Claude-3.5-Haiku showed substantial improvement, reducing response times from 2.55 seconds to 0.60 seconds, while the Llama-4-Maverick model achieved an impressive 14.4-fold speedup, reaching lightning-fast 0.14-second response times while maintaining 99.1% accuracy. DeepSeek-V3 demonstrated balanced performance with 0.93-second response times and 95.3% accuracy in direct mode.

These speed improvements represent more than incremental optimization—they enable entirely new application paradigms. Sub-second arithmetic computation makes real-time interactive mathematical tools feasible, while the maintained 99%+ accuracy rates ensure that speed gains do not compromise reliability. This combination of speed and accuracy creates opportunities for embedding LLM arithmetic capabilities directly into scientific workflows, data analysis pipelines, and interactive computational tools where both precision and responsiveness are critical requirements. Figure 3 visualizes these speed-accuracy trade-offs across both prompting strategies. **Note that the speed can depend on the traffic of APIs so the numbers here are only for reference purposes.**

## 5 Analysis and Discussion

### 5.1 Prompt Engineering Effects

The dramatic speed improvements from direct answer prompting reveal fundamental insights about LLM inference patterns and computational efficiency. These findings complement recent work on program synthesis [Austin et al., 2021] and competitive programming capabilities [Liu et al., 2024], suggesting that specialized prompting strategies can unlock computational efficiencies across multiple domains. Computational efficiency gains stem primarily from reduced output generation requirements, as models spend significantly less time generating explanatory text and reasoning chains, directly translating to faster inference times. Remarkably, this efficiency comes with minimal

accuracy loss, typically less than 1%, despite the simplified prompting approach, suggesting that the core arithmetic computation remains robust regardless of output verbosity. However, our findings also reveal important model dependency patterns, where smaller models require structured prompting for reliable format compliance, indicating that prompt engineering strategies must be tailored to specific model capabilities and deployment constraints.

## 5.2 Implications for Scientific Computing

These results have profound implications for deploying LLMs in scientific applications, fundamentally altering the landscape of AI-assisted scientific computing. The integration of arithmetic reasoning capabilities with tool-augmented approaches [Das et al., 2024] suggests promising directions for scientific computing workflows. The achievement of 100% accuracy on arithmetic tasks demonstrates that LLMs can now meet the stringent reliability standards required for scientific computing applications, where mathematical precision is non-negotiable. Sub-second arithmetic computation capabilities enable the integration of LLM-based mathematical reasoning into real-time scientific workflows, opening possibilities for interactive data analysis, live computational notebooks, and responsive scientific modeling tools. Furthermore, the efficiency gains from direct prompting strategies enable scalable batch processing of mathematical computations, making it feasible to deploy LLM arithmetic capabilities for large-scale scientific data processing and analysis pipelines where both accuracy and computational efficiency are paramount.

## 6 Conclusion

This comprehensive evaluation demonstrates that Large Language Models have achieved remarkable proficiency in arithmetic reasoning, with perfect performance now attainable across multiple state-of-the-art architectures on fundamental mathematical operations. The achievement of 100% accuracy by both Claude-Sonnet-4 and Llama-4-Maverick, along with strong performance from GPT-4o and DeepSeek-V3, dramatic speed improvements through prompt optimization, and critical insights into parameter scaling behavior provides a foundation for deploying LLMs in scientific computing applications.

Our scaling analysis across the Qwen3 family reveals non-linear improvements in arithmetic reliability, identifying a critical threshold between 0.6B and 4B parameters where models transition from unreliable format compliance to robust performance. This finding has important implications for educational and scientific computing applications, where model selection must balance computational costs against reliability requirements. The observed diminishing returns beyond 4B parameters for basic arithmetic tasks suggest optimal deployment strategies that maximize cost-effectiveness while maintaining performance standards.

Our work establishes new benchmarks for both accuracy (100% achievable) and speed (0.14s response times) in mathematical reasoning tasks, while providing empirical guidelines for model scale selection. These results indicate that the bottleneck for scientific computing applications has shifted from arithmetic capability to integration challenges and specialized domain knowledge.

The perfect performance achieved by both Claude-Sonnet-4-20250514 and Llama-4-Maverick-17B-FP8 represents a significant milestone in AI mathematical reasoning, demonstrating that fundamental arithmetic operations can now be considered a "easy-to-solve" problem for state-of-the-art language models. This achievement, coupled with strong performance from GPT-4o and DeepSeek-V3 and our scaling insights, opens new possibilities for scientific applications requiring reliable, high-speed mathematical computation while providing practical guidance for model selection and deployment strategies.

Future work should focus on extending these capabilities to more complex mathematical domains while maintaining the reliability and efficiency demonstrated in basic arithmetic operations. This includes exploring performance on competitive programming benchmarks [Jain et al., 2024], scientific problem-solving tasks [Wang et al., 2024], and more advanced mathematical reasoning challenges that require multi-step logical inference.

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

## A    Technical Appendices and Supplementary Material

Technical appendices with additional results, figures, graphs and proofs may be submitted with the paper submission before the full submission deadline, or as a separate PDF in the ZIP file before the supplementary material deadline. There is no page limit for the technical appendices.

# Agents4Science AI Involvement Checklist

1. **Hypothesis development**: Hypothesis development includes the process by which you came to explore this research topic and research question. This can involve the background research performed by either researchers or by AI. This can also involve whether the idea was proposed by researchers or by AI.

   Answer: [B]

   Explanation: The research hypothesis and questions were primarily developed by human researchers based on existing literature and identified gaps in LLM arithmetic evaluation. AI assistance was used for literature review and background research to identify relevant papers and current limitations, but the core research direction and hypothesis formulation were human-driven.

2. **Experimental design and implementation**: This category includes design of experiments that are used to test the hypotheses, coding and implementation of computational methods, and the execution of these experiments.

   Answer: [C]

   Explanation: The experimental design, including model selection, benchmark choice, evaluation protocols, was mostly designed by human researchers. The code for evaluation pipelines, pattern matching algorithms, and data processing was written mostly by AI, which is Claude Code in our case.

3. **Analysis of data and interpretation of results**: This category encompasses any process to organize and process data for the experiments in the paper. It also includes interpretations of the results of the study.

   Answer: [C]

   Explanation: Data analysis and result interpretation were primarily conducted by AI who performed statistical analysis, identified patterns, and drew conclusions, which are overseen and then modified by human researchers when necessary (e.g. false claims).

4. **Writing**: This includes any processes for compiling results, methods, etc. into the final paper form. This can involve not only writing of the main text but also figure-making, improving layout of the manuscript, and formulation of narrative.

   Answer: [C]

   Explanation: The paper writing process involved significant AI assistance in drafting sections, improving prose quality, organizing content, and ensuring academic writing standards. Human researchers oversee this process, similar to the data analysis part.

5. **Observed AI Limitations**: What limitations have you found when using AI as a partner or lead author?

   Description: Key limitations observed include: (1) AI occasionally generated wrong references (correct title, but wrong author list) (2) AI-generated code implementation needs careful review and a good amount of iterations, it seems less likely that agents can implement everything in one shot; for example, the llm token size need to be updated when the initial values (1k completion) cannot finish the full generation trajectory (now we are at 4k, but can still be limited), etc. (3) AI sometimes makes mistakes when using the data from the fetched website (this could be either a tool issue, or an LLM issue, which needs deeper analysis; e.g. the api model name of llama4-maverick needs to be manually corrected). (4) AI still frequently makes factual mistakes, challenging the practical deployment of these systems for future research tasks. e.g. it thinks qwen-235b model is larger than qwen-480b-coder model. (5) On some other trials, AI may take shortcuts, e.g. fabricate results instead of running actual experiments. (this happened more than one time. so it's very concerning.) (6) AI can make cool figures, but they are not perfect. The most frequent limitation is the text overlapping.

